# Precision Imaging Guidance in the Era of Precision Oncology: An Update of Imaging Tools for Interventional Procedures

**DOI:** 10.3390/jcm11144028

**Published:** 2022-07-12

**Authors:** Chiara Floridi, Michaela Cellina, Giovanni Irmici, Alessandra Bruno, Nicolo’ Rossini, Alessandra Borgheresi, Andrea Agostini, Federico Bruno, Francesco Arrigoni, Antonio Arrichiello, Roberto Candelari, Antonio Barile, Gianpaolo Carrafiello, Andrea Giovagnoni

**Affiliations:** 1Department of Clinical, Special and Dental Sciences, University Politecnica delle Marche, 60126 Ancona, Italy; alessandrabruno92@gmail.com (A.B.); nicolorossini44@gmail.com (N.R.); a.agostini@staff.univpm.it (A.A.); a.giovagnoni@univpm.it (A.G.); 2Division of Special and Pediatric Radiology, Department of Radiology, University Hospital “Umberto I—Lancisi—Salesi”, 60126 Ancona, Italy; a.borgheresi@staff.univpm.it; 3Division of Interventional Radiology, Department of Radiological Sciences, University Politecnica Delle Marche, 60126 Ancona, Italy; r.candeari@ospedaliriuniti.marche.it; 4Radiology Department, Fatebenefratelli Hospital, ASST Fatebenefratelli Sacco, 20122 Milan, Italy; michaela.cellina@asst-fbf-sacco.it; 5Post-Graduation School in Radiodiagnostics, Università degli Studi di Milano, 20122 Milan, Italy; giovanni.irmici@unimi.it (G.I.); arrichielloantonio@gmail.com (A.A.); 6Department of Biotechnological and Applied Clinical Sciences, University of L’Aquila, 67100 L’Aquila, Italy; federico.bruno.1988@gmail.com (F.B.); antonio.barile@univaq.it (A.B.); 7Emergency and Interventional Radiology, San Salvatore Hospital, 67100 L’Aquila, Italy; arrigoni.francesco@gmail.com; 8Operative Unit of Radiology, Fondazione IRCCS Ca’ Granda Ospedale Maggiore Policlinico di Milano, 20122 Milan, Italy; gianpaolo.carrafiello@unimi.it; 9Department of Health Sciences, Università degli Studi di Milano, 20122 Milan, Italy

**Keywords:** interventional radiology, imaging guidance, oncological therapy, embolization, biopsy, percutaneous treatments

## Abstract

Interventional oncology (IO) procedures have become extremely popular in interventional radiology (IR) and play an essential role in the diagnosis, treatment, and supportive care of oncologic patients through new and safe procedures. IR procedures can be divided into two main groups: vascular and non-vascular. Vascular approaches are mainly based on embolization and concomitant injection of chemotherapeutics directly into the tumor-feeding vessels. Percutaneous approaches are a type of non-vascular procedures and include percutaneous image-guided biopsies and different ablation techniques with radiofrequency, microwaves, cryoablation, and focused ultrasound. The use of these techniques requires precise imaging pretreatment planning and guidance that can be provided through different imaging techniques: ultrasound, computed tomography, cone-beam computed tomography, and magnetic resonance. These imaging modalities can be used alone or in combination, thanks to fusion imaging, to further improve the confidence of the operators and the efficacy and safety of the procedures. This article aims is to provide an overview of the available IO procedures based on clinical imaging guidance to develop a targeted and optimal approach to cancer patients.

## 1. Introduction

Interventional oncology (IO) procedures have become extremely popular in interventional radiology (IR) and play an essential role in the diagnosis, treatment, and supportive care of oncologic patients through new and safe procedures based on different image techniques guidance [1,2], especially during the current COVID-19 pandemic [3]. Interventional radiologists conduct a wide spectrum of percutaneous and transcatheter endovascular cancer therapies that offer effective clinical benefits and are easily distinguishable from medical, surgical, and radiation oncological treatments, defining IO as the “fourth pillar of cancer care” [4].

The continuous development of IR allows for optimal management of many malignancies, avoiding the need for open invasive surgical interventions, with a reduced number of complications, faster recoveries, cheaper costs, and clinical success [5].

IO can be divided in four branches [6]: namely IR, interventional radiotherapy, consisting of treatments and procedures involving the use of radioisotopes (i.e., brachytherapy and transarterial radioembolization [7,8]; interventional chemotherapy, based on focused chemotherapy such as transarterial chemoembolization; and interventional endoscopy, involving treatments and procedures including an endoscopic approach, such as percutaneous gastrostomy, catheters, and stents positioning.

The goal of IO is to offer personalized treatments allowing for the elimination of the tumor, with minimal injury to the adjacent healthy tissues. To do that, interdisciplinarity and highly specialized teams play a significant role in the diagnostic workout, treatment decision and planning, and procedure performance and effectiveness [6]. Each treatment should be planned based on the characteristics of the individual patient and the engagement of experts in multiple disciplines.

IO, offering high safety profiles and efficacy through minimally invasive procedures, also has a central role in the management of elderly and fragile patients who often have complex comorbidities precluding them from more invasive options for cancer treatment or who can benefit from palliation of cancer-related symptoms. Bone ablation, for example, has proven to be an effective therapy for painful bone metastases, representing a valid alternative to radiation therapy when contraindicated [9,10].

This article aims at providing an overview of the available IO procedures based on clinical imaging guidance to develop a targeted and optimal approach to cancer patients.

## 2. Vascular and Non-Vascular Approaches

IR can nowadays rely on a great variety of effective and safe treatments in the oncologic field that can be divided into two big groups: the vascular [11,12,13,14] and the non-vascular treatments [15].

Endovascular approaches can be chosen in the management of different lesions and are based on embolization and local administration of chemotherapy.

The embolization decreases lesion vascularization and is used as a preoperative procedure to reduce the lesion size and the risk of bleeding during surgery; the concomitant injection of the chemotherapeutic agents during the embolization increases the effectiveness of the drugs and increases the treatment efficacy.

Endovascular catheter-directed therapies may be used to treat direct tumor targets. Transarterial locoregional therapies include different options, such as bland transarterial embolization, transarterial chemoembolization, and selective internal radiation therapy (SIRT). Bland transarterial embolization and chemoembolization are the two predominant endovascular approaches, while the third one is represented by transarterial radioembolization, which delivers radioactive yttrium-90 through a catheter in the hepatic artery [16].

Bland transarterial embolization involves inserting a catheter into the hepatic artery branches supplying the lesion and then obstructing them with an embolic agent to achieve a total arterial occlusion of the lesion vessels. Gelatin (slurry or pledgets), nonabsorbable particulate agents (polyvinyl alcohol or PVA), and trisacryl gelatin microspheres are some of the embolic agents that can be used.

Transient elevations in transaminases and bilirubin levels as well as an association of symptoms defined as “postembolization syndrome” [17] are possible side effects of the procedure. Abdominal pain, malaise, nausea, and low-grade fever are common symptoms that are typically self-limiting; severe complications such as liver and renal failure, gastric ulcers, and death (2–3%) have been recorded in very small percentages; non-target embolization is uncommon [18].

Transarterial chemoembolization (TACE) is based on the simultaneous use of chemotherapy and embolic agents and aims at delivering a high concentration of a chemotherapeutic drug directly into the arterial blood supply of the tumor [19]. TACE should be as selective as possible; the injection of embolic particles into the hepatic artery branches feeding the tumor determines significant tumor necrosis [20,21]; the intravascular co-injection of chemotherapeutic agents results in a local concentration and permanence of the drug, which limits the systemic diffusion and toxicity of the treatments, and enables a targeted and effective lesion treatment [21]. The most common complications are self-limiting, but serious complications can manifest in 2–3% and include hepatic and renal failure, ulceration, and death [22].

Transarterial radioembolization with yttrium-90 (90Y) microspheres is a form of brachytherapy that uses the hepatic arterial supply to permanently implant 90Y in the target tumor through the injection of resin or glass microspheres [23].

The mechanism of action of transarterial radioembolization is that 90Y microspheres lodge inside the tumor capillary and release damaging radiation, achieving local doses of 100–1000 Gy over a small area of approximately 2.5 mm of tissue penetration for a limited period. Tumor blood flow allows producing reactive oxygen species when water molecules ionize because of radiation interactions: increased reactive oxygen species development triggers cellular oxidative stress in tumor cells, which contributes to apoptosis activation. This technique aims to avoid radiation exposure to healthy tissues; therefore, an accurate pre-procedural assessment of the lesion supply is needed [24].

Erroneous non-target vessel radioembolization results in various complications ranging from skin irritation through the falciform artery to more serious issues, such as ischemia and necrosis of the stomach, gallbladder, and small bowel; liver and kidney failure, gastroduodenal ulceration, and death have been also reported in small numbers [22].

Percutaneous approaches include different procedures such as percutaneous image-guided biopsies and different ablation techniques, with radiofrequency, microwaves, cryoablation, and focused ultrasound (US). The invasiveness of these approaches is limited, as the target lesions are reached through needles, and open surgery is avoided.

Percutaneous biopsy is a minimally invasive alternative to surgical biopsies. It can be applied in many tumors with different image guidance techniques (US, magnetic resonance imaging (MRI), computed tomography (CT)). Image-guided biopsy is often the pre-chemotherapy preferred biopsy method and can be associated with IR procedures such as post-biopsy tract embolization with gelatin foam or thrombin to minimize the risk of post-procedural hemorrhage [25].

Radiofrequency ablation (RFA), microwave ablation, and cryoablation are the most popular thermal ablation approach applied in clinical practice. Moreover, irreversible electroporation and high-intensity focused ultrasound (HIFU) are recent technologies that are playing an increasing role in the treatment of different tumors [26]. RFA aims to deliver thermal energy into tissue, resulting in the coagulation necrosis of the tissues close to the probe tip. The too-quick heating of the tissue next to the probe tip causes desiccation, increasing the tissue impedance and limiting the propagation of energy to surrounding molecules and the ablation area [27].

Technological development of probes design improved the size of the ablation zone, but the burn zones are still smaller than what other ablation technologies would achieve [28]. Another limitation is represented by the susceptibility to the heat-sink effect, which is the dissipation of energy next to the burn zone caused by flowing blood [28].

Microwave ablation uses energy that is at the higher border of the radiofrequency spectrum. Cell death caused by microwave ablation is nearly identical to that produced by RFA. Microwave energy shows some advantages over RFA: the microwave ablation induces molecular oscillation and thermal energy delivery in tissues more distant from the probe site, obtaining wider, hotter ablation zones more quickly than RFA [29].

Cryoablation-induced cell death is caused by the Joules–Thomson effect, in which a pressurized gas (argon) rapidly decompresses, resulting in a dramatic temperature drop and cytotoxic temperatures of −25 °C or less [30]. Freezing causes the development of intracellular and extracellular crystals, organelle death, and membrane destruction. The two major strengths of cryoablation are the less susceptibility to the heat-sink phenomenon and the real-time visualization of the ablation zone with imaging.

A rare complication is the “cryoshock” phenomenon, caused by a release of intracellular elements and cytokines into the systemic circulation, resulting in hypotension, tachycardia, thrombocytopenia, disseminated intravascular coagulation, and multi-organ failure, and has been reported in association with the ablation of large lesions [31]. The problem of post-treatment hemorrhage has been solved with the development of newer probes allowing for tract cauterization.

Irreversible electroporation is based on a strong, short-pulsed electromagnetic field that is applied to the tissue and causes the trans-membrane potential to increase. Membrane permeability becomes irreversible when the transmembrane potential exceeds a certain threshold, resulting in cell apoptosis and necrosis [32]. This approach is not affected by the heat sink and tissue desiccation and provides greater geometric predictability for ablation and allows the sparing of the connective tissue architecture, with consequent minimal risk of damage to the adjacent healthy structures.

HIFU allows ablative necrosis without the need for needles and incisions through a US beam that delivers thermal energy passes through the tissue without damaging it, resulting in minimal invasiveness [33].

These procedures are considered very safe, as, in addition to the optimal control in releasing energy, several systems to control sensitive structures are available: thermocouples to track the temperature reached near sensitive structures or the ability to transfer sensitive structures away from the ablation area using air, carbon dioxide, or sulfur dioxide injection [5].

This technique is unfortunately affected by some disadvantages: first, the patient positioning and thermal energy delivery can last up to three hours; second, the respiratory motion of the treated organs, despite respiratory-gating applications, can affect the treatment effectiveness. Moreover, interposed bone structures can create an impenetrable barrier to ultrasound energy.

## 3. Imaging Planning and Guidance

Target IO treatments consist of multi-phase procedures based on the application of different clinical imaging modalities.

The phases of treatment planning can be divided into pre-procedural imaging planning, intraprocedural lesion targeting, procedural monitoring, post-procedural assessment of treatment effectiveness, and potential complication and imaging follow-up [34].

The pre-procedural imaging planning should be performed with high-quality images to assess the indication to the procedure, feasibility, technical best approach, the lesion site and size and vascularization, feeding vessels, eventual risk represented by anatomical variations, and neighboring structures [35].

Intraprocedural lesion targeting can be performed through different imaging techniques, and real-time modalities are usually preferred. US is the real-time modality for excellence; CT is considered a near-real-time modality. Other non-real-time imaging examinations can also be applied to best guide the interventional procedures alone or combined in fusion-images approaches [36,37].

The monitoring of the procedure is essential to assess if the treatment has been localized correctly through the modification of the treated target [38].

Post-procedural assessment is essential to evaluate the treatment effectiveness and completeness and potential complication [39]. Follow-up imaging has the role to monitor remaining post-procedural viable tissue or relapse, usually analyzing the presence of contrast enhancement and dimensional variations over time. In Figure 1, the steps of imaging guidance procedures are summarized.

## 4. Imaging Modalities

### 4.1. US

US represents a powerful tool in IR [40,41] and may be the most used guidance technique. It shows several benefits: first of all, the wide availability, the limited costs, and the possibility to be performed at the patient’s bed, which is particularly important in critically ill patients in the intensive care unit who cannot be moved to the IR department. It provides real-time guidance during any procedure [42], visualizing the course of needles or catheters for biopsies and interventional procedures. Other main advantages are the absence of ionizing radiation, which makes it a suitable procedure also for pediatric patients and pregnant women, and the reduced required procedural time when compared to CT-guided approaches.

It has excellent anatomical detail, which can be further implemented with the association of US applications, such as color Doppler, contrast-enhanced US (CEUS), and elastography [43].

Many types of IR procedures can be performed with US guidance; the most common is tissue biopsy to diagnose lesions nature and obtain tissue for molecular tests and cytogenesis [44], ablation procedures, thoracentesis, paracentesis, drainage positioning in abscesses of parenchymal organs, peritoneal abscesses/fluid collection, and empyema [45].

Other applications consist of guidance for percutaneous transhepatic cholangiography and nephrostomy [46], guidance for supportive care in cancer patients (e.g., gastrostomy), guidance for vascular access, and tumors complication treatments [47,48], and for ablation procedures, CEUS is an excellent alternative to traditional imaging techniques and, in some cases, may even exceed other modalities for image-guided procedures, according to growing evidence in the literature [49].

It is based on the use of contrast agents consisting of microbubbles with a diameter ranging from 2 to 6 mm, which are tiny enough to move through capillary beds but too wide to get into the interstitial space, functioning as acoustic enhancers. They can be administered intravenously, evaluate vascularity and perfusion, intracavitary, into physiological or pathological cavities to determine the cavity morphology and its communications with neighboring structures and organs, or in the association.

As the conventional US, CEUS enables a real-time visualization with an excellent temporal resolution useful for procedural guidance and also to assess rapid variations in the contrast-enhanced characteristics: lesion/tissue contrast enhancement can be checked constantly, with a perfect visualization of wash in and wash out, without the radiation exposure which characterizes CT, avoiding the missing a delayed contrast extravasation or enhancement anomalies.

CEUS represents a useful tool in different IO procedures: it improves the visualization of lesions to guide targeted percutaneous biopsies of tumors poorly or not visible in the conventional US and, particularly, of their vascularized portions, to collect samples of the vital tumor areas, and allow the identification of post-biopsy bleedings [50].

Second, it provides accurate guidance for ablation treatments such as RFA, microwave ablation, and cryotherapy, allowing a precise positioning of the probe into the lesions when poorly evaluable on the conventional US, and is a readily usable tool for treatment effectiveness assessment [51], as the persistence of post-ablation residual vascularity indicates the need for further ablation and detection of hemorrhagic complications [2]. CEUS can also be used for lesion follow-up to avoid the administration of any nephrotoxic contrast media and radiation exposure.

Third, this imaging technique can guide IR procedures essential for the supportive care of cancer patients, such as percutaneous nephrostomy by increasing the visibility of the calyceal system, and represents the perfect follow-up imaging to check the right positioning of biliary and pleural drainage catheters and possible post-procedural complications, such as bleedings or vascular fistulas [11,49]. CEUS limitations are mainly in common with conventional US ones and consist of patients’ physique and high operator dependence.

Another advanced US application that can improve image-guided IO is represented by elastography [52,53]: this rapidly evolving US technology provides information on tissues’ mechanical characteristics, such as their hardness or stiffness. This information can be used to plan liver- and renal-targeted biopsy to select the most suitable tissue/lesion regions and avoid areas at higher risk of post-biopsy bleedings [41].

Images from US, CT, and MRI can now be displayed concurrently and in real-time, depending on the angle of the transducer, thanks to developments in imaging technology, combining the nonionizing radiation US guidance with the information from the cross-sectional contrast-enhanced imaging. Fusion CT/MRI–US imaging increases target lesion visibility and aids in understanding the three-dimensional relationship between the lesion and the adjacent anatomical structures [36,54].

The fusion approach increases the operator confidence, the accuracy of the targeting, and the technical success, as it is characterized by several strengths: first, it can increase the detectability of small lesions, particularly in the case of poor conspicuity on the B-mode US. Second, it allows the precise insertion of therapeutic needles within the tumor (Figure 2). Third, CT/MR–US fusion image guidance can decrease the number of treatment sessions.

Even though the masses are not observed in US with a based system such as GPS in US systems, their projection on the US is determined by detecting them in CT and MRI, and biopsy can be easily performed in US thanks to fusion [51].

### 4.2. CT

In the last decade, the role of CT in oncologic imaging shifted from an exclusively diagnostic tool to an integrated one of the interventional suites. CT provides advantages in terms of spatial resolution; however, there are some challenges to consider: it provides fixed bidimensional images, and therefore, cases where an “off-axis approach” of the device insertion is necessary are more challenging [55], and in such cases it does not allow “real-time” images. Different CT approaches can be applied to guide IO procedures.

The main added value of CT in interventional radiology is the excellent spatial resolution [47], which allows a confident execution of complex ablation, biopsies, or drainages. Commonly, a grid or the CT gantry laser light localizes the entrance for the target lesion; however, as it does not allow real-time visualization of the needle, it still relies on the experience of the interventional radiologist. This fact may result in multiple steps for the needle positioning and tracking with a consequent increase of the procedure time and radiation dose to the staff and the patient and, eventually, of periprocedural complications (Figure 3).

The navigational guidance tools could potentially overcome these issues [56]. The main navigational tools applied in CT-guided procedures are tracking systems based on an electromagnetic or optical laser method and robotic guidance systems [57] that trace the movement of the instruments in real-time based on an electromagnetic or optical system: the probe is placed manually and guided through continuous tracking of the instrument. Robotic systems do not provide real-time visualization of the probe but are calibrated to the CT scanner and provide active guidance of the ablation probe in robot-assisted navigation [58]. These navigational tools also take advantage of images fusion, where images from different imaging modalities are co-registered and overlay it; this technique is applied to biopsies [59], ablation on vascular or nonvascular procedures, and to combine the modalities to obtain a better real-time spatial visualization of the needle or the target lesion [60,61]. In some cases, it is possible also to perform a triple overlay of the images [62]. The tracking systems work as “body-GPS”, merging in a three-dimensional space the real-time “tracking coordinate system” and the procedural planning “image coordinate system”.

The “tracking coordinate system” includes an electromagnetic or optical detector and fiducial markers placed on the patient’s skin and the instruments and allows to instantly calculate the position of the probe in a three-dimensional space. The “image coordinate system” is not in real-time, as it is obtained from previously acquired CT/MRI images used for the procedural planning, and to compensate for breathing movement, it could be integrated with respiratory gating of the fiducials placed on the patient [63]. However, to obtain a “true-real time” image, real-time US can be co-registered to previously acquired imaging modalities that better visualize the target lesion. Tracking systems showed promising results and have been validated mostly for RFA, MWA [64], and cryotherapy, showing an increased accuracy in needle placing, fewer needle repositionings, primary efficacy [65,66], and lower radiation dose [67,68].

Robotic systems predefine the entry point, angle, and depth of the ablation probe, providing active guidance during the procedure; the main strength of this approach is the reduced probe repositioning, particularly in procedures with out-of-plane targets [69].

The main concern of CT guidance is the radiation dose to the interventional radiologist and the patients [70] even if some studies reported values in line with recommended dose limits for occupational radiation exposure [71]. To reduce the time needed for placing the probe/needle and therefore the radiation dose, some navigation systems have been developed to assist lung biopsies and thermal ablation therapies, and studies demonstrated that CT assisted with stereotactic navigation reduces the time for ablation probe placement, the number of needle adjustments, skin punctures, and fluoroscopy time [72,73]. The disadvantages of CT-guided procedures are the lack of tumor visibility on unenhanced images or, in the case of contrast media administration, the short contrast-enhancement timeframe [74,75].

#### CT Angiography (CTA)

CTA was introduced in the 1990s to unify CT and angiography [76]. One of the main fields of application of angio-CT is in the study and treatment of focal liver lesions. This integrated system allows to perform CT during hepatic arteriography when the catheter is placed in the hepatic artery or celiac trunk and CT during arterial portography, when the catheter is placed in the superior mesenteric or splenic artery [77]. CT examination could be performed before the procedure to provide better visualization of the target lesion and further simultaneous lesions [78]. For liver tumor ablation, angio-CT allows for repeated administration of small doses of intra-arterial intrahepatic contrast medium, improving tumor conspicuity and precision of the needle placement [79].

CTA has several advantages in transarterial treatments, as it provides three-dimensional vascular images of the tumor-feeding arteries to be embolized, including extrahepatic-feeding arteries [80], with easier and more accurate identification and embolization of the feeding arteries [81]. In patients affected by hepatocellular carcinoma (HCC), this allows performing super-selective catheterization of the subsegmental feeding arteries reducing the damage to the surrounding healthy tissue and the amount of anticancer and embolizing agent provided, with improvements in survival rate.

The combination of an accurate vascular anatomy and perfused-liver volume calculation for dosimetry provides a significant added value in transarterial radioembolization, where non-target embolization would have severe consequences [82]. CTA can also be repeated during or immediately after the procedure to assess the technical success, the complications, and, eventually, refine the results of the treatment. This is valuable for performing in the same session combined treatments (transarterial and ablation) of selected unresectable HCCs [83].

When compared to cone-beam computed tomography (CBCT), CTA demonstrated lower radiation dose [82,84,85] and reduction of the contrast media administration [84]; moreover, it provides a wider field of view and less image noise [86,87]. However, CTA is not widely available when compared to other imaging methods; it is more expensive and requires greater room spaces.

### 4.3. CBCT

CBCT is an imaging technique consisting of a rotating C-arm equipped with a flat panel detector that provides fluoroscopic, digital subtraction angiographic, and volumetric CT images in a single-patient setup [88]. The different imaging modalities, 3D and 2D images, can be combined and co-displayed so that each one overcomes the defects of the other in planning, monitoring, or verifying the results of the treatments.

There are numerous proposals for the best acquisition technique and contrast media administration protocol, which mainly rely on the specific clinical task (planning, monitoring, and verification) [89].

In the last decade, the technical improvements reduced the rotation time of the C-arm, which has become short enough to obtain in one contrast injection two sequential arterial acquisitions, namely an early “arteriographic” arterial phase and a late “lesion” arterial phase, providing after a single contrast injection the information of both the lesion feeding vessels and the parenchymal staining [90]. Moreover, the development of software and reconstruction algorithms for integration of the 3D and 2D images allows building a personalized treatment strategy for each case of percutaneous or intra-arterial treatment. 

The availability of volumetric datasets showing tumor location, vascularity, and surrounding tissues during the procedure provides an excellent base to establish a safe and effective route to the target and guide device positioning.

One of the main advantages of CBCT over CT in the setting of percutaneous treatments is the lack of a small and deeply located gantry and the greater flexibility in the orientation of the detector around the patient.

The first task while performing a transcutaneous treatment is the visibility of the target lesion. In this sense, CBCT can provide considerable advantages over other more widely available imaging modalities (i.e., CT, US), for example, in lung nodules and liver lesions not visible in the US. The second task is to select the optimal route for reaching the target lesion. CBCT provides “CT-like images” that may result particularly useful in cases where the path to the target lesion requires medial/lateral and caudal/cranial angulations and intraprocedural adjustments. 

CBCT paired with virtual navigation systems allows performing biopsies, ablations, and percutaneous transthoracic localization of pulmonary nodules too small or faint to be detected for the surgeon [91].

The biopsy of lesions poorly visible on CT or close to anatomical structures poorly visible on CT (e.g., nerves or vessels) could also represent a challenging procedure with CBCT, and when it is possible, US or MRI guidance should be preferred [92].

#### Fusion Technique

US guidance represents the most used technique for liver ablation treatments. However, when the probes are placed under US guidance, the concomitant use of CBCT detects the need for probes repositioning in 73% of cases [93]; moreover, not all liver lesions are visible (Figure 4). On the “CT-like” images of the CBCT, dedicated software can automatically calculate the safe electrode path, and then, electrode deployment can be performed under the US or real-time fluoroscope, with images reaching good results even in lesions ≤ 1.5 cm [94] (Figure 5). US/CBCT fusion represents a useful tool to increase the correct targeting of poorly US-visible HCC nodules in the angio-suite [95,96].

The navigation software can also virtually plan the treatment volume determining the number of antennas required to achieve complete tumor coverage [93].

A recently developed software can co-register CBCT images of the post-treatment with pre-treatment CT to enhance the presence of residual tumor [97] (Figure 6).

In addition to the high spatial resolution, the main added value of CBCT while performing endovascular treatments is the capability to enhance the vascular tree over the digital subtraction angiography while providing a three-dimensional road map. This is particularly valuable in cases where the vascular anatomy is altered by the underlying pathology, as in the cirrhotic liver.

The main role of CBCT in transcatheter arterial treatments for HCC is in terms of tumor detection, feeding vessel identification, and vessel navigation.

CBCT has a tumor detection rate of 90% [98], with sensitivity increased in proportion to tumor size [99] and vascularization [99,100] and could identify small angiographically occult tumors allowing super-selective catheterization [100].

The first software for tumor vessel identification on CBCT images was introduced more than a decade ago [101]. Since then, automated or semi-automated software for the tumor-feeding arteries was developed and is widely used [102,103,104], with an excellent detection rate of the arterial feeding vessels [105] (Figure 7).

It also should be taken into consideration that HCC could have an extrahepatic arterial supply, which has been reported in about 17–27% of cases [106]. The identification of extrahepatic feeding vessels is mandatory to ensure a complete treatment and to avoid hemorrhagic complications [98,107].

The added value of CBCT for transarterial liver treatment is clear in terms of tumor and feeding vessel detection; however, it is not completely clear if this may result in improving tumor response, and several efforts are made in this direction [108]. An interesting new development is virtual liver perfusion mapping, which is a technique that allows estimating virtual vascular territories after positioning a virtual injection point on nonselective dual-phase CBCT images [109]. This technique is demonstrated to provide reliable images to evaluate the technical success of transarterial treatments [110,111].

CBCT can be also used for SIRT for a catheter-directed treatment approach to treat primary and secondary liver tumors with yttrium-90-loaded microsphere infusion into the hepatic artery due to its accuracy in the identification of perfused tissues, allowing a correct lesions segmentation and possible pre-treatment portal vein embolization.

CBCT guide embolization can also be used for the treatment of renal lesions as a pre-surgical procedure to prevent hemorrhagic complications in highly vascularized renal cell carcinomas and to reduce the hemorrhagic risk in angiomyolipoma [88].

Combining treatments with TACE and local ablation treatments (RFA or MWA) demonstrated to improve the clinical prognosis of the patients for HCC and renal-cell carcinoma [112].

CBCT is a versatile tool allowing the performance of both procedures confidently (Figure 8). A few studies on the combined treatment of HCC and both MWA or RFA with TACE suggest that it improves the clinical outcome [113]: in particular, when compared to TACE alone, it demonstrated a longer progression-free survival with comparable complication rates [113]. Promising results were also recorded in a case series of T1a renal-cell carcinomas treated with MWA and renal artery embolization [112]. However, further studies are needed.

### 4.4. MRI

#### 4.4.1. MRI Guidance

The main incentives for investing in research in MRI rather than the other techniques working with X-rays (CT in particular) were represented by (a) lack of radiation exposure; (b) superior contrast resolution (without using contrast medium); and (c) ability to demonstrate the temperature changes. Even though the low doses required today by modern CT scanners and angiographic equipment have partially reduced the appeal of MRI as a guiding technique during interventional procedures, recent research confirms the advantages (b) and (c) [114,115]. Moreover, the development of larger and shorter gantries eases the interventional approach. On the other hand, specific instrumentation is required to operate within the MRI suite, not to mention the need for operators’ skills and knowledge during the procedures (e.g., radiofrequency generators may interfere with MRI and should be inactivated during the scanning process). In fact, MRI-guided procedures are not widely diffused because of the need for MR-compatible devices, limited availability of interventional MR suites, and the longer times required for multiple MR sequences, but the fusion of MRI with CBCT images could overcome all these issues, combining the advantages of the modalities in a standard interventional suite [92].

Biopsies and ablations are the most common procedures performed under MRI guidance [116]. The main advantage is a better depiction of the target without contrast medium administration [117]. Real-time multiplanar imaging combined with high-contrast resolution allows accurate localization of lesions poorly defined on the unenhanced CT; moreover, during the ablation procedures, MRI allows the monitoring of the needle pathway and the complete visualization of the tissue ablated both in case of heat-based techniques thanks to temperature-sensitive sequences and in the case of cold-based ones [118,119] (Figure 9).

Another advantage is the possibility to continuously (even with double obliquity, typical of US guidance), coupled with thermal control, evaluate the area of ablation.

The image guidance for ablations is typically represented by MRI, providing the anatomical information for the right beam targeting and temperature mapping.

Multiple applications have been developed targeting both visceral organs and soft tissue lesions. Kidneys represent suitable targets both for biopsy and ablation [118,119]. MRI is excellent for the detection and staging of prostate lesions [120]. Biopsy of the prostate can be performed fusing US and MRI [121] or directly performing in-bore biopsy with the advantages of real-time MRI to confirm needle position, to eliminate misregistration or organ movement affecting fusion imaging, and to reduce the bacterial infection risk using trans-perineal instead of the transrectal pathway. Multiple cases of ablation with favorable results in terms of safety and complications have been described; the main issue remains the indication even though both focal and whole gland ablations are safely and effectively performed [114]. According to the experiences describing liver ablations, MRI guidance is possible [122], but the US approach appears more feasible [123].

Given the growing role of MRI in screening, surgical planning, and follow-up, indications for MRI-guided biopsy procedures are also increasing [124,125]. Biopsy under MRI guidance should be performed in any case where other imaging methods—in particular ultrasound—are not sufficient to clearly visualize the lesion to be biopsied [126]. The cancer detection rate of MR-guided breast biopsy, though with differences among studies, is reported up to 50% [124,127]. Particular attention should be paid to the phase of the menstrual cycle and in patients under hormone-replacement therapy due to the known effects of the background parenchymal enhancement (BPE) affecting the MRI signal. In cases where biopsy cannot be technically feasible, MRI guidance can help the pre-surgical lesion localization using guide wire or marker clip placement [125].

In summary, MRI seems to be a better guidance technique when US and CT offer poor conspicuity, when a reduced exposure in terms of radiation and contrast is required, and when, due to challenging locations, a complex pathway requires continuous control of the needle [119].

#### 4.4.2. MRgFUS

MRI-guided focused US surgery (MRgFUS) is a particular procedure in which high-intensity focalized US are fused with MRI. The system is highly integrated, as the thermal ablation system is engineered to work within the MRI and offers the great advantage of the continuous thermal and imaging control during energy administration in order to have real-time feedback of the effects of the heat during energy delivery, thus avoiding definitive and undesired lesions. One of the most remarkable fields of application of MRgFUS is the ablation of the deep nuclei of the brain (i.e., the ventral intermedium of the thalamus, VIM) for the management of the essential tremor and the Parkinson-related tremor [33].

Bone is another well-documented field of application [128,129]. Benign lesions and bone metastases on the bone surface are safely and effectively treated (Figure 10). Uterine fibroids and adenomyosis foci are also treated with this technique [130].

Several studies have evaluated the application of HIFU—primarily through ultrasound guidance but also through MRI guidance—in pancreatic cancers either for palliative treatment or combined with other therapies (chemotherapy, radiotherapy, surgery) [131]. Although follow-up timing and outcome measures differ between studies, most authors report a variable tumor response (in terms of devascularization and/or tumor size reduction), while a common endpoint across all studies is the significant effect of HIFU treatment in the palliation of pain [132].

## 5. New Frontiers

The interest in artificial intelligence (AI) in radiology is fast-growing not only in diagnostic imaging but also in interventional radiology [133]. Deep learning algorithms showed promising results in diagnostic imaging in helping the most effective customization of treatments and triage of resources [134] and can also be applied in IO to predict the efficacy of therapy, improving patients’ selection.

AI can be applied in different phases of IO: for the outcome prediction and patients’ selections and as a supporting tool for clinical and imaging decisions, for a pre-procedural accurate imaging assessment and image quality improvement, and as intra-procedural assistance (e.g., selection of the most appropriate materials). Precision medicine relies on the concept of better patient selection. AI decision-support systems may help tailor treatment decisions based on imaging phenotypes, yielding better clinical results [135]. Interventional radiologists often rely on multidisciplinary boards for oncological treatment strategies. These board discussions perform multiparametric risk-stratification, integrating the patients’ data before treatment is advised; different AI applications replicate and outperform these discussions by predicting the outcomes and/or the benefits of treatment before performing it [136].

The ability to join clinical and imaging pre-procedural data, radiomics [137,138], and genetic information may improve the precision of decision making and be a helpful tool in risk assessment, patient triage, and outcome prediction [139].

Applications of AI are also available in the procedural phase: deep learning systems can improve the image quality of digital subtraction angiography through the correction of translational motion via pixel shifting or by training a neural network to predict the subtracted images from the unsubtracted images [140].

The synchronization of preoperative cross-sectional imaging with intraoperative real-time fluoroscopy or US provides through automatic multi-modality registration and segmentation tools [141] results in more precise guidance for biopsies and local image-guided therapies and enhances problem-solving abilities throughout procedures [93].

A virtual angiogram or “angioscopy” can be generated using CT or MRI data to guide endovascular procedures [142], and real-time identification of the most suitable guidewire, catheters, and stents to treat stenotic vascular lesions or aortic aneurysms is also available [143].

Furthermore, the use of novel navigational techniques such as augmented reality and virtual reality that superimpose virtual pre-procedural 3D anatomic data onto real-world 2D visual images in real-time can conceptualize complex anatomies, converting DICOM images into 3D models, allowing the simulation of complex situations through 3D cameras and the consultation of virtual data (clinical and imaging) in the interventional suite, and providing holograms handle on the procedural field, thus improving accuracy for minimally invasive treatments and reducing risks, complications, and radiation exposure [144,145].

## 6. Conclusions

The era of personalized medicine presents a great opportunity for cancer imaging and treatment. Minimally invasive techniques for the diagnosis, treatment, and palliation of cancer provided by IO are protagonists of innovative modern medicine, and IO specialists should be considered essential players in the multidisciplinary management of oncologic patients. The role of IO procedures is continuously increasing and will continue to grow in the future to allow even more targeted and customized patient treatments.

## Figures and Tables

**Figure 1 jcm-11-04028-f001:**
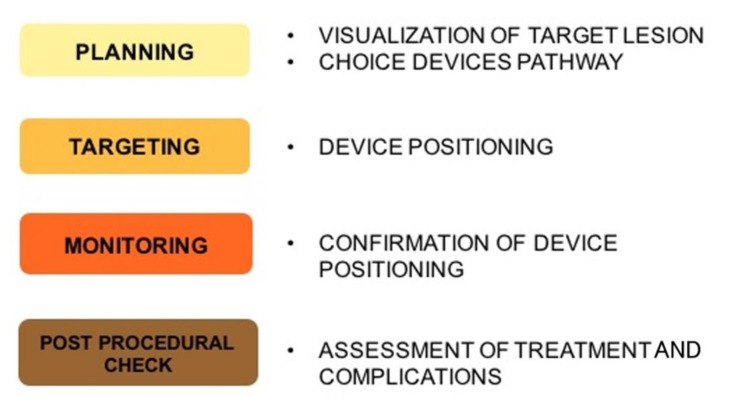
Imaging guidance steps: Imaging is at the core of minimally invasive approach, and selection of the optimal diagnostic and interventional modality is critical for a safe and effective treatment delivery.

**Figure 2 jcm-11-04028-f002:**
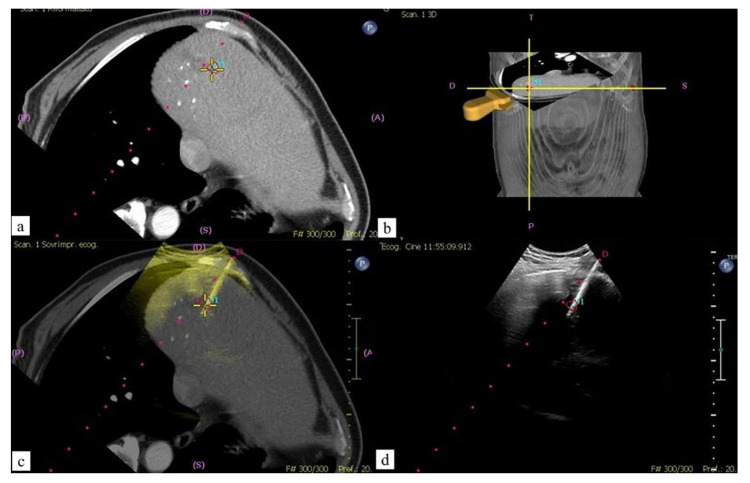
Sixty-four-year-old male patient with CRC liver single metastasis: percutaneous MWA. (**a**,**b**) Contrast enhanced CT (CECT) post-processed images with ablation planning; (**c**) fused images obtained using CECT and real time US during MW probe positioning; (**d**) US imaging confirmation.

**Figure 3 jcm-11-04028-f003:**
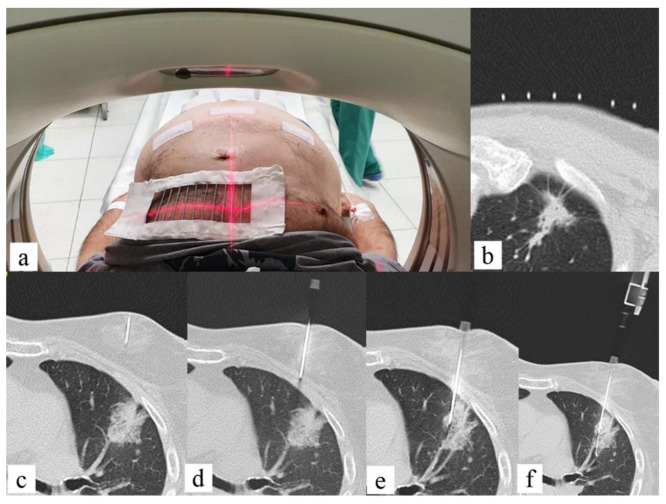
Fifty-six-year-old male with lung lesion: CT guided biopsy. (**a**) Patient is positioned supine depending on the lesion location; the region of interest on the patient thoracic skin is delimitated with a radiopaque grid; (**b**) the entry point on the skin is signed on basis of coordinates given by the intersection of grid markers and CT level; (**c**) the anesthesia needle is left on the skin, and a CT scan is performed to check the correct needle trajectory previously determined; (**d**–**f**) the anesthesia needle is replaced with a 20 G coaxial needle that is advanced from the entry point to the target point on basis of CT images planning with a “step-by-step” CT guidance technique.

**Figure 4 jcm-11-04028-f004:**
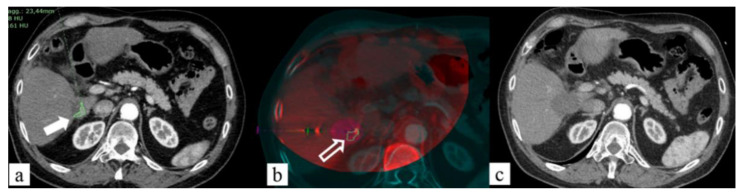
Sixty-seven-year-old male, residual tumor (arrows) close to inferior vena cava margin after US-guided RFA of HCC: MWA ablation. (**a**) Contrast-enhanced CT (CECT) axial image with segmentation of residual tumor to treat (fill white arrow); (**b**) fused images obtained using CECT and CBCT after US-guided MWA probe positioning into residual tumor (empty white arrow) with predicted ablation area overlaying and the segmented tumor visualized on CT image; (**c**) 1-month follow-up CECT demonstrated complete ablation of residual tumor treated (no residual hypervascular area).

**Figure 5 jcm-11-04028-f005:**
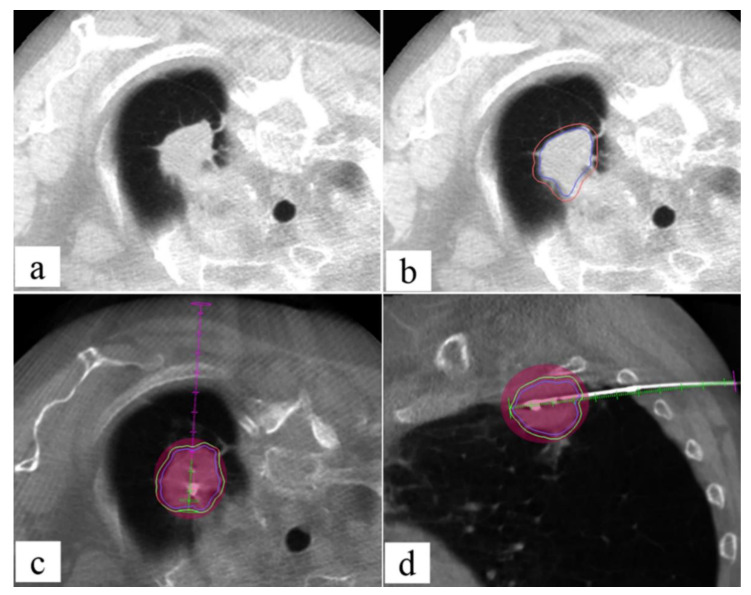
Seventy-five-year-old female, pulmonary adenocarcinoma: MWA ablation. (**a**) Pre-procedural CBCT: lesion visualization; (**b**) with lesion segmentation (blue line) and adding 5 mm safe margins (orange line); (**c**) ablation planning: MW virtual probe with predicted ablation area (manufacturer specific) (pink area) overlaying to segmented lesion (blue line) with 5 mm safe margins (green line) covered in axial plane; (**d**) intra-procedural CBCT: MW probe placement with all lesions with 5 mm safe margins (green line) covered by the predictable ablation area in sagittal plane.

**Figure 6 jcm-11-04028-f006:**
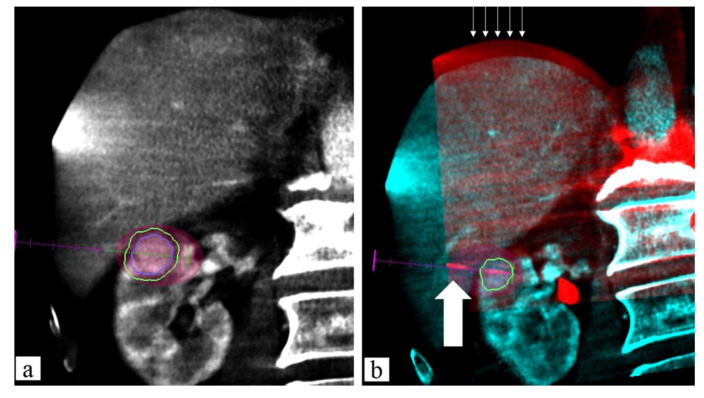
Eighty-three-year-old male, renal clear cell carcinoma biopsy-proven: MWA ablation. (**a**) Contrast-enhanced (c.e.) pre-procedural CBCT planning: lesion segmentation (blue line) and adding 5 mm safe margins (green line) with MW virtual probe with predicted ablation area (manufacturer specific) overlaying to segment; (**b**) fused images obtained using c.e. preprocedural CBCT and non-enhanced intra-procedural CBCT: MW probe (white wide arrow) placement with all lesions covered by the predictable ablation area. Thin white arrows show the mismatch overlay on the diaphragm plane between the two CBCT images due to different breath holding.

**Figure 7 jcm-11-04028-f007:**
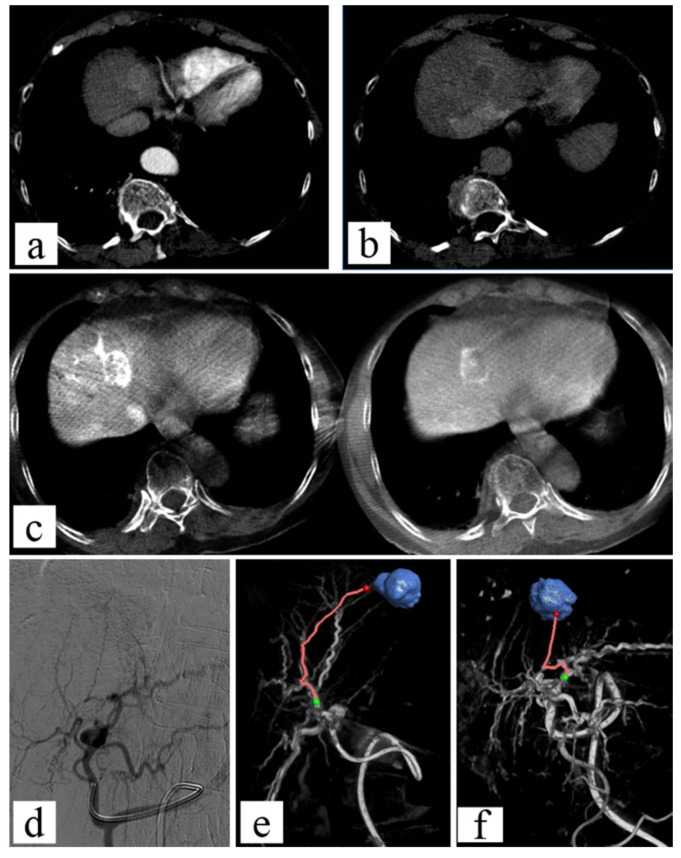
Seventy-year-old patient suffering from HCC: TACE. (**a**,**b**) Diagnostic contrast-enhanced CT in arterial and portal phase; (**c**,**d**) dual phase constrast-enhanced CBCT imaging shows the disisomogeneous enhancement of the lesion; (**d**) angiographic imaging does not demonstrate the lesion visualization; (**e**,**f**) using Emboguide software on CBCT imaging, the tumor feeders are identified.

**Figure 8 jcm-11-04028-f008:**
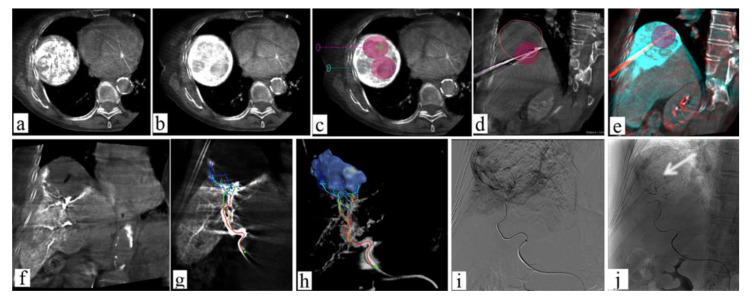
Seventy-eight-year-old female with very large HCC: combined therapy with MWA and TACE. (**a**,**b**) Large HCC lesion visualized with CBCT; (**c**) planning CBCT for ablation of HCC lesion, with volume ablation prediction; (**d**,**e**) fused images obtained using c.e. preprocedural CBCT and non-enhanced intra-procedural CBCT (red lines show large HCC lesion); (**f**) post-ablation contrast-enhanced CBCT with evidence of the ablation area and peripheral ipervascular tumor residual; (**g**,**h**) using Emboguide softwaere on CBCT imaging, the tumor feeders are identified; (**i**) angiographic images of microcatheter advancement on arterial feeders; (**j**) angiographic image after lipiodol-TACE of residual tumor demonstrates the lipiodol pooling on residual tumor (arrow).

**Figure 9 jcm-11-04028-f009:**
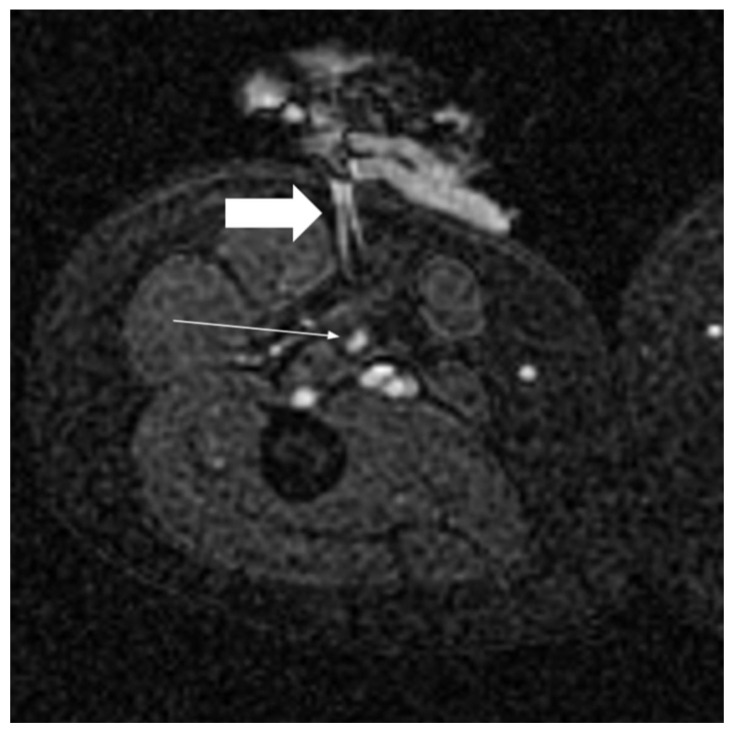
Image from a MR-guided cryoablation of a small relapse of soft-tissue tumor (thin white arrow); the MR-compatible needle for cryoablation (thick white arrow) with MR artifact.

**Figure 10 jcm-11-04028-f010:**
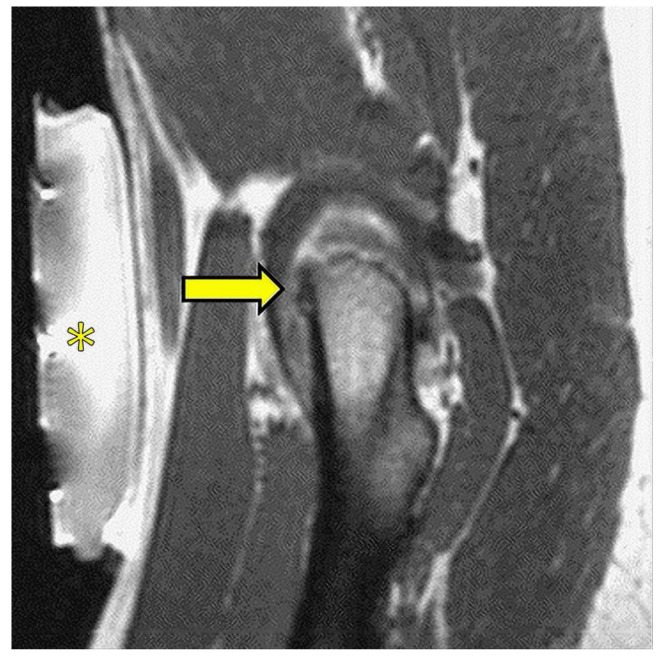
Image from a MRgFUS treatment of a small osteoid osteoma of the femoral neck (arrow); the transducer (*) lies on the patient’s skin and produces and focuses the ultrasound beam.

## Data Availability

Not applicable.

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
