# Peer review of "Precision Imaging Guidance in the Era of Precision Oncology: An Update of Imaging Tools for Interventional Procedures"

_jcm, 2022, doi:10.3390/jcm11144028_

Round 1
Reviewer 1 Report
The authors have presented a paper about "Precision Imaging Guidance in the era of Precision Oncology: an update of imaging tools for interventional procedures".
The topic is absolutely interesting however I believe that in the introduction it should be assessed more clearly the clinical landscape of this evolving field; in fact Interventional Oncology does not only include inteventional radiology but several other braches (such as interventional radiotherapy, interventional chemotherapy and interventional endoscopy). Please read doi: 10.5505/tjo.2019.4 for rurther details.
Another important point which should be highlighted regards the role of the multidisciplinary approach in the choice of the procedures: the authors should underline his relevant aspect.
A third point which in my view deserves more attention within the paper is the possible use of these approaches in elderly and frail patients: in fact locoregional interventional procedures have this huge advantage which should emerge more clearly from the discussion.
Reviewer 2 Report
Thanks to the author(s) for the review. The points that need to be edited are listed below.
On page 4, the sentence starts with 'the image guidance..' disrupts the flow of the article, it should be placed in the section about MR.
In paragraph 7 on page 4, the word fluoroscopy for CT- fluoroscopy should be removed and CT fluoroscopy statements in the entire article should be replaced with 'CT'. Because both methods are combined. Separate nomenclature is not required.
On page 5, instead of the 4.US heading, imaging modalities should be written.
Instead of 4.US, start with A.US and continue like this.
Supplementary materials 1 written on page 5 could not be reached.
The sentence begins with 'This section…' at the end of the 3rd paragraph on page 6 could not be understood.
The paragraph starts with 'The most recent ...' on page 6. should be removed.
6. The Fusion Imaging US on the page should get the title number or letter again. In addition, the Fusion Imaging US section should be added to the US section in a more concise form.
The first sentence of paragraph 4 on page 6 is not understood.
6. The last two paragraphs on the page should be deleted.
An explanation can be added here: "Even though the masses are not observed in US with based systems such as GPS in US systems, their projection on the US is determined by detecting them in CT and MRI, and biopsy can be easily performed in US thanks to fusion (reference should be given)".
CT should be placed on this page because of the developmental stage on page 7.
On page 7, the letter X in the word xray in the first paragraph should be capitalized.
On page 8, procedures in breast masses should be added to the interventional procedures related to MRI with reference.
On page 9, MRgFUS procedures in pancreatic masses should be mentioned with reference.
In Figure 4, the section where the mass is seen should be added and the mass should be shown with an arrow.
The sentence starting with 'CT-fluoroscopy' on page 11 should be removed.
The sentence starting with 'the main guidance' on page 11 disrupts the flow and should be removed.
The paragraph starting with 'the main concern' on page 11 should continue with a separate paragraph from this section.
On page 11, CT Angiography should be renamed as a subtitle.
The CBCT may not be given as a separate heading but as a separate modality heading. Because it forms the main part of the article.
In the second paragraph on page 12, it is implied that contrast examination is performed only in CBCT. It should be removed or dynamic CT applications with contrast should be added here. For example, the same applications can be made in dual CT.
The paragraph starting with The main applications on page 12 should be deleted, this information has already been given.
On page 12, in the CBCT percutaneous treatments biopsies section;
The first three paragraphs should be added to the previous paragraphs. The next three paragraphs should be deleted.
Paragraphs 1-3 and 4 on page 13 should be deleted.
A sentence that begins with 'The biopsy of lesions' is not entirely valid. It must be rewritten.
The place of the sentence that starts with However is not in this section. It should change location.
On the page 13 in section 8.2;
A separate section in the form of Fusion technique should be opened and this information should be added there.
The paragraph starting with If the lesion should be placed in the contrast examinations section.
8.3 does not need to be given under a separate heading on page 15.
Paragraph 2 on page 16 should be deleted.
The first paragraph on page 17 contains information unrelated to the article.
Section 8.42 does not need separate headers
The sentence given by reference 160 on page 18 is not needed.
The conclusion section should be added at the end of the article.
As a result, interruptions to the flow of thought in the article make it difficult to read. It should be re-evaluated after major revision.
